# Future pHealth Ecosystem-Holistic View on Privacy and Trust

**DOI:** 10.3390/jpm13071048

**Published:** 2023-06-26

**Authors:** Pekka Ruotsalainen, Bernd Blobel

**Affiliations:** 1Faculty of Information Technology and Communication Sciences (ITC), Tampere University, 33100 Tampere, Finland; 2Medical Faculty, University of Regensburg, 93053 Regensburg, Germany; bernd.blobel@klink.uni-regensburg.de

**Keywords:** privacy, trust, holistic view, fiducial duty, privacy law, smart contract

## Abstract

Modern pHealth is an emerging approach to collecting and using personal health information (PHI) for personalized healthcare and personalized health management. For its products and services, it deploys advanced technologies such as sensors, actuators, computers, mobile phones, etc. Researchers have shown that today’s networked information systems, such as pHealth ecosystems, miss appropriate privacy solutions, and trust is only an illusion. In the future, the situation will be even more challenging because pHealth ecosystems will be highly distributed, dynamic, increasingly autonomous, and multi-stakeholder, with the ability to monitor the person’s regular life, movements, emotions, and health-related behavior in real time. In this paper, the authors demonstrate that privacy and trust in ecosystems are system-level problems that need a holistic, system-focused solution. To make future pHealth ethically acceptable, privacy-enabled, and trustworthy, the authors have developed a conceptual five-level privacy and trust model as well as a formula that describes the impact of privacy and trust factors on the level of privacy and trust. Furthermore, the authors have analyzed privacy and trust challenges and possible solutions at each level of the model. Based on the analysis performed, a proposal for future ethically acceptable, trustworthy, and privacy-enabled pHealth is developed. The solution combines privacy as personal property and trust as legally binding fiducial duty approaches and uses a blockchain-based smart contract agreement to store people’s privacy and trust requirements and service providers’ promises.

## 1. Introduction

Nowadays, we live in almost borderless digital environments where products are increasingly interchangeable with intellectual and informational goods and services [1]. They also require the availability of “Big Data” related to human’s experiences, relationships, behaviors, and environments. Novel health service models such as tele-health, mHealth, pHealth, eHealth, and digital health are examples of those services. pHealth focuses on personal/personalized health and health care services, and it presents a horizontal view of health care, eHealth, and mHealth [2]. Modern pHealth services are data-driven and vary in focus and size. The pHealth service can be a sensor system that is focused on a dedicated personal health or wellness problem, or it can be a personal health recommender system using a holistic view of a person’s health [3]. Nowadays, pHealth services are increasingly part of a dynamic, multi-stakeholder, cross-organizational, cross-border, and cross-jurisdictional ecosystem. In pHealth, technological innovations have always been adapted on the front lines. Currently, the deployment of smart sensors, mobile devices, wireless networks, web technologies, digitalized services, Cloud platforms, algorithms, artificial intelligence (AI), machine learning (ML), and blockchains for online personalized health services is common. Ongoing paradigm shifts in health care from organization-centered and reactive healthcare to person-centered preventive and predictive care are well adapted in pHealth [4].

Today’s information technology enables Web sites, computer applications, and networks to routinely collect, use, store, and share all kinds of personally identifiable information (PII) about a person’s health problems, including health-related information about the person’s life. Modern sensors, wearables, and smart wrists have the ability to measure a person’s physical activity, blood pressure, heart rate, quality of sleep, social activities, stress, emotions, and mood [5]. Furthermore, behavioral activities are invisibly tracked online when using computers, mobile phones, and health services via networks [6,7]. According to Zuboff, almost unlimited data collection and surveillance are daily practices in the digital age [8]. Video surveillance systems in public spaces can monitor our social and health-related behaviors. Data analytics companies both sell our raw data and use our PII in the form of behavioral profiles and predictive products [9].

The Internet of Things (IoT) and artificial intelligence of things interfaces (AIoT) are new emerging technologies that enable real-time data collection. According to Ziegeldorf et Al., the IoT moves the collection of personal data and behaviors from the internet and public spaces to homes and working places [10]. The novel neurotechnology has the ability to go even further. According to Berger et al., it can impact technology indirectly through wearable devices that read data from the head and also write data using neuromodulation. According to Berger et al., in the future, neurotechnology may have the ability to influence people’s behavior, emotions, values, and thoughts [11].

During the last few years, both the public sector and private organizations have shown increasing interest in the collection and use of personal health data for innovations, new products, and services, and they have built technology environments such as ecosystems where services offered are dependent on the collection of PHI, users’ behaviors, and interactions [12]. This development has raised the question of the ownership of PHI and whether or not health data should be understood as a public good (a commodity produced without profit for all) or personal property [13]. Currently, there is no unanimous answer to this question. According to Piasecki et al., the ownership concept cannot solve problems associated with the sharing of PHI [14]. In a workshop report, Crossmann et al. summarize that health care data should be established as a public good [15]. Taylor sees that nowadays, data as a public good model fits best with corporate reality and existing models for data sharing [16]. On the other side, propertization of personal information, according to Schwarz, responds best to people’s concerns about privacy [17]. The new European Health Data Space proposal goes even further by considering health data according to the common good model. In this proposal, health data is understood to include not only EHR and clinical trials but also the content of PHR and personal wellness data [18].

These days, the Internet brokers and data giants have well understood the commercial value of health data and the potential of AI and have adapted and established the business model of data collection and commercialization [19]. Increasing commercialization of PHI is a big problem for human rights, and it raises the danger that, in the future, expected economic benefits will override people’s needs for privacy and autonomy. It is notable that the United Nations has confirmed that privacy remains a human right even in the digital age, and “sharing health data as a public good requires making data available with the right degree of openness or restriction to achieve maximum benefit, while reducing any potential for harms” [20]. It is evident that privacy is a big problem in real-life networks and ecosystems [21]. Therefore, information privacy is inevitable in the digital age [22]. Researchers have shown that traditional security-based privacy protection solutions cannot guarantee privacy, and a person cannot control the collection, use, and sharing of his or her personal information (PII) in today’s networked information systems [2].

Building trust in a dynamic ecosystem is a big challenge for the service user. In today’s information systems, it is widely expected that people blindly trust organizations’ and service providers’ promises that they process PII fairly. In other words, it is expected that a service user believes without any proof that structures such as guarantees, regulations, promises, legal recourse, and procedures are in place (i.e., structural assurance) and that the environment is in proper order (i.e., situational normality) [23]. Unfortunately, researchers have shown that in real life, this is far from true. And in dynamic multi-stakeholder ecosystems, it is almost impossible to know who and why to trust [24]. Furthermore, researchers have observed that digital information systems are seldom designed with privacy in mind, i.e., in today’s digital information systems, trust is only an illusion [25,26]. As pHealth services are built over the same general ITC technology used in commercial information systems and platforms, they share the same privacy and trust concerns [26,27,28,29]. It is a specific feature of pHealth ecosystems that some of the stakeholders can be non-regulated health care providers or private organizations. This means that parts of personal health information (PHI) collected and used are not regulated by health care-specific laws. Together with the sensitivity of PHI collected and used, this raises additional privacy concerns, especially because the content and implementation of privacy laws vary in different countries [30].

The future of pHealth has the potential to offer personalized, preventive, and predictive services to its service users. However, for it to be successful, it needs a huge amount of PHI and health-related personal behavioral data covering a person’s regular life, social relations, economic activities, and psychological status [2]. This data (i.e., personal big health data) can be used for different analyses, to calculate detailed personal health profiles, to detect changes in personal health and disease, and to develop new applications such as personal health recommendation services [3]. According to researchers, future pHealth will rely on Internet of Things (IoT)-based data collection and advanced computer methods such as machine learning (ML), artificial intelligence (AI), and deep learning (DL) [27,29,30]. It is evident that in the future, pHealth privacy and trust challenges will be much bigger than they are today. To realize even a part of the promises of data-driven pHealth (e.g., innovations, new products, better health, and economic growth) and to prevent short- and long-term negative consequences for human values such as privacy, dignity, integrity, and autonomy, it is necessary to find new solutions for privacy, trust, and ownership of PHI.

This paper is an extension of the work originally presented to the pHealth 2022 Conference [2]. In that paper, the authors have studied methods and solutions that have the ability to prevent the situation where PHI can be invisible collected, shared, and misused, where there is no information privacy, and where predefined trust in technology and service providers’ fairness is just expected [22,27]. As healthcare-specific laws and general privacy regulations grant several rights to the data subject (e.g., access to their own health data, rectification, objection, data portability, and the right to block data sharing), this paper is focused on the collection, processing, storage, and sharing of PHI that takes place outside the health care domain and medical research [31]. The authors’ starting point is that future pHealth should be ethically acceptable, trustworthy, and empower the service user or data subject (DS) to maintain information privacy by expressing their own privacy needs and expectations. For future pHealth, potential ICT solutions and their weaknesses are discussed, and the authors also propose a holistic set of principles and solutions that, when used together, have the power to make future pHealth ethically acceptable and trustworthy.

The rest of the article is organized as follows: Chapter 2 briefly summarizes the main features of widely used privacy and trust models and the principles of information ethics. In chapter 3, the authors define how the pHealth ecosystem is understood in this paper and present a user’s view of it. In Chapter 4, privacy and trust challenges existing in current pHealth ecosystems are discussed. Then (Chapter 5), features of new privacy and trust approaches developed by researchers are analyzed. In chapter 6, a five-level holistic model and a formula describing factors that influence the level of privacy and trust in an ecosystem are presented. In Chapter 7, the authors propose a holistic solution for a trustworthy, privacy-enabled, and ethically acceptable pHealth ecosystem. Chapter 8 covers the limitations of this paper and outlines the necessary future steps needed to reach the authors’ goal.

## 2. Privacy, Trust and Information Ethics

Privacy and trust are vague, dynamic, situational, and context-dependent concepts with many definitions [2,27,32]. Almost all cultures value privacy, but they differ in how they obtain it [33]. It is widely accepted that privacy is a human and constitutional right [34]. Information privacy is a subset of the concept of privacy [35]. According to Floridi, two theories of information privacy are popular: the reductionist interpretation and the ownership-based interpretation. The first theory looks for undesirable consequences caused by the misuse of data, and the second theory defines that a person owns his or her information (privacy is defined in terms of intellectual property) [36]. According to Smith et al., the person who owns PII can also trade privacy for other goods or services [37]. For Decew, privacy is a common value because all individuals value some amount of privacy [33].

At a general level, privacy addresses the question “what would we like others to know about us”. In western countries, privacy is widely based on concepts of autonomy and informational self-determination, which refer to a person’s right and expected ability to control the flow of his/her own personal information. This implies that a person has the right to control when, by whom, and why personal information is collected and shared, and to protect himself/herself against surveillance, unnecessary data collection and processing, dissemination, unauthorized use, and harm caused by the unfair use of PII [38,39,40].

Other widely used privacy models are privacy as a concern, legal construct, risk-based concept, behavioural concept, and social good [27,41,42]. The concept of privacy as a commodity understands privacy as an economic good that can be traded. The privacy as a concern approach refers to individuals’ anxiety regarding the collection, processing, and unfair use and sharing of data. Privacy as a regulative (legal) construct tries to regulate the way data is collected, used, and shared [43]. The risk-based approach to privacy focuses on risks such as harm caused by unnecessary data collection, misuse, disclosure, surveillance, and behavioral manipulation [44,45].

In real life, the nature of privacy and the lack of availability of reliable privacy related information make the measurement of the actual (objective) level of privacy challenging [45]. Furthermore, researchers have found that privacy preferences vary drastically from individual to individual. They can change over time and are context-dependent [46]. Furthermore, in many countries, privacy is not an absolute right. Instead, it can be balanced with, or overridden by, others’ concerns and priorities, including business needs, public safety, and national security. According to Friedewald, the right to privacy requires a forward-looking privacy framework that positively outlines the parameters of privacy in order to prevent intrusions [22]. A meaningful challenge is that, while technical solutions provide some protection against data misuse, the existence of such protection does not necessarily mean that users will disclose more information [47].

Some researchers have pointed out that current privacy models do not work in distributed and digital information systems, and there is a need to redefine how we understand privacy [21,48]. Furthermore, Friedewald has proposed that in the digital age, the concept of privacy should be expanded to include the following aspects: privacy of the person, privacy of personal behavior and actions, privacy of personal communication, privacy of data and images, privacy of thoughts and feelings, privacy of location and space, and privacy of association (including group privacy) [22].

According to Sætra, an individualistic model of privacy is insufficient, and privacy should be understood as a public good, i.e., everyone in a society should have the right to enjoy privacy [49]. DaCosta has proposed a novel privacy-as-property approach. Its fundamental idea is that “you have the right to control yourself, and this property interest in oneself extends to the external objects you own, including your data” [50]. Acquisti et al. have proposed that in the digital age, privacy should include not only personal data but also behaviors and actions, personal communication, thoughts and feelings, and associations [51].

Traditionally, trust is understood to exist between persons; however, researchers agree that trust is also needed between a person and an organization (organizational trust) and between a person and technology (trust in technology) [29]. Trust is needed in situations where the trustor has insufficient information about the features and behavior of the trustee [52]. Therefore, to build a relationship of trust, there must be confidence that the other partner will act in a predictable manner [39]. Trust can also be defined as a personal expectation of other partners’ future behavior [53]. Trust is widely understood as a disposition, attitude, belief, feeling, expectancy, psychological state, personal trait, social norm, subjective feature, willingness to be vulnerable, and perception based on one’s own previous experiences or others’ recommendations. Trust has been understood as the willingness to depend on other parties expected or unexpected actions without the ability to monitor or control them [54]. Perceived trust is a personal opinion based on information gleaned from one’s own senses or from others. Computational trust is an algorithmic imitation of human-based measured features of the trustor and the used information system. Thus, trust is often based on emotions or feelings, which also include cognition. According to Ikeda, trust can be based on justifiable reasons such as laws and science [55]. That kind of trust, aka “rational trust” is based on rational arguments [56]. In the case of rational trust, the trustor should have facts on which the trust is based [57].

In digital information systems, people should increasingly trust technology. According to Mc Knight, trust in technology is often a belief that the technology used is reliable, secure, and protects information privacy, and that appropriate governance is established and enforced [58]. Furthermore, the level of trust depends on the understanding of the system and its behavior, i.e., the system’s willingness and ability to correctly perform [59].

Ethics is a set of principles and concepts that judge whether a behavior is right or wrong. The basic principles of general ethics are autonomy, justice, non-maleficence, privacy, and solidarity. Normative theories of ethics include consequence-based theories, duty-based theories, rights-based theories, and virtue-based theories to guide how to interact properly with others [60]. Information ethics (which is closely related to computer ethics) is an applied ethics that focuses on the relationship between the creation, organization, dissemination, and use of information and the ethical standards and moral codes governing human conduct in society [61]. Information ethics intertwines with other areas of applied ethics such as computer ethics, data ethics, internet ethics, engineering ethics, and business ethics. In this paper, the authors emphasize that information ethics covers not only humans but also any actor in the ecosystem, such as applications and technologies, including implantable and wearable devices. As today’s networked information systems, or more generally, ecosystems, impact human values such as life, health, happiness, freedom, knowledge, resources, power, and opportunity in many ways, researchers have highlighted that information systems should function in an ethically acceptable way [62]. The European Union has proposed the following ethical principles for information systems using AI: The system must not negatively affect human autonomy, violate the right to privacy, or directly or indirectly cause social or environmental harm to an individual. Instead, the system should support freedom and dignity. Furthermore, the AI system should be accountable and transparent to its stakeholders and end-users [63]. The authors state the above-discussed ethical principles as mandatory for all information systems processing PHI.

## 3. User View on Privacy and Trust in pHealth Ecosystems

The concept “ecosystem” was originally developed in the fields of ecology and biology [64]. Today, it is transferred to many other contexts and widely used in the field of information science to describe networked communities consisting of interconnected and interrelated technical and non-tangible elements [54]. Typical non-tangible elements are data, digital services, and stakeholders. Technical elements include networks, platforms, programs, and communication lines. Architecture presents its structure, function, and relations [64]. The goal of the ecosystem is to create value for all stakeholders [64]. A pHealth ecosystem is typically a socio-technical system that is characterized by its stakeholders’ business models, roles, and relations, its services and products, information flows, the information itself, and the underlying infrastructure [65]. Nowadays, pHealth ecosystems are increasingly platform ecosystems. In the pHealth ecosystem, there can be conflicting objectives; e.g., stakeholders want to maximize their profit by collecting and using a maximal amount of PHI, and users try to minimize short- and long-term harms through disclosed data while at the same time getting benefits from services. In pHealth ecosystems, a service user typically has a direct connection to one service provider. On the other hand, other parts of the ecosystem, including its other stakeholders, architecture, deployed privacy technology, regulations, business goals, and relations, are usually invisible to the service user, i.e., the ecosystem looks at the service user as a black box (Figure 1)**.**

This means that the service user is strongly dependent on the service provider’s willingness to use and share the service user’s PHI ethically and fairly. The service user cannot build trust in the pHealth ecosystem and make privacy-based decisions without sufficient information about the ecosystem’s invisible features, which the service provider should make available. According to Dobkin, service providers who collect and utilize user data are fiduciaries to customers [66], and the information fiduciary duty of service providers could ensure that they use data only in ways that are consistent with users’ expectations [67]. This implies that there exists a specific informational and fiducial relationship between the service user and service provider in pHealth ecosystems. Thereby, the service provider has an obligation to act in the best interests of the service user.

## 4. Privacy and Trust Challenges in pHealth Ecosystems

When we currently use information systems, we leave trails that expose our interests and traits that expose our actions, beliefs, intentions, and targets of interests to commercial entities and also to our governments [51]. On the Web, there are tens of thousands of health apps collecting and using PHI [32]. In today’s information systems, a person’s behavioral health data is systematically and secretly collected by Web service providers, health apps, and Web platforms. According to Dobkin et al., at least 77.4% of Websites globally track visitors’ data and people’s behavior [66]. Zuboff has noted that behavioral tracking is not only used to measure body signals but also our physical and social behaviors and how we use information systems [8]. Therefore, persons do not have sufficient knowledge of what information other people, organizations, and firms have about them, by whom and how that information is used, and what the consequences are for the data subject (DS). This situation is often described by decision-makers and firms as a win-win situation, i.e., people, organizations, firms, and society benefit from data sharing. Unfortunately, in real life, benefits to a person are often only promises or beliefs, and invisible data collection and sharing not only breaches trust and privacy, but it can also generate harm to the person.

Researchers have shown that traditional privacy solutions such as notification and choice (consent) as well as fair information processing principles have failed to guarantee privacy in today’s networked environment. Although the new privacy regulations, such as the EU-GDPR, oblige companies to specify how the collected data is used, in real life, organizations’ privacy policies (if available) are written in a legalistic and confusing manner and are difficult to understand and use [68]. For the same reasons, transparency in the form of an organization’s privacy policy does not work well. Furthermore, many big Web actors and service providers simply do not care about privacy laws [45,69]. Finally, the service user’s privacy decision takes place in a complex, multidimensional situation, and his or her bounded rationality makes rational privacy choices difficult [70].

Currently, most Web services are free of charge for the consumer, but in real life, people are paying for the use of services through disclosed PII (i.e., PII is traded and monetarized). The option of buying an application with the option of not disclosing personal information to the company, which may subsequently sell that data, does not exist [68]. Service providers and platform managers often expect that they can freely use and sell disclosed data [2]. Service providers are also prioritizing their business needs and benefits over service users privacy needs. There is also a tension between public and commercial interests in collecting and using PII on the one hand and people’s needs for privacy on the other. Industry widely sees personal information as raw material for products and services and society as a public good. This makes it challenging to balance a person’s individual need for privacy on the one hand and the use of PII for meaningful public benefits or for making profit on the other [27].

The unlimited collection of a person’s behavioral data is a big problem in today’s digital information systems. Behavioral data talks about our routines, habits, and medical conditions. Behavioral data can also be used to uniquely identify individuals [71] and web-browsing behavior [72]. Another problem is that in ecosystems, the DS cannot know how data will be used in the future, what its potential uses are, or whether other people’s PHI will be linked to it [73].

Lack of trust is also a meaningful problem in ecosystems, where the user has to trust not only the service provider but also other frequently unknown stakeholders and surrounding information technology. In ecosystems, the service user does not have reasonable knowledge of stakeholders’ trust features and relations and has no power to negotiate privacy rules and safeguards or force the service provider or platform manager to take personal privacy and trust needs into account [2]. Instead, the service user is typically forced to accept a service provider’s privacy promises (policy) and trust manifesto in the form of a take-it-or-leave-it approach [66]. Unfortunately, commercial service providers often have low incentives to enforce strong privacy policies, and they often do not keep the privacy promises expressed in their policy documents [73,74]. This all indicates that policy-makers and technology firms fail to provide the user with reasons to trust, and codes of conduct and privacy policies will not provide sufficient reasons to trust [75].

As discussed earlier, the vagueness of privacy and trust concepts makes it difficult to conceptualize and measure them and to make them understandable for computer programs. To solve this problem, different proxies such as service level agreements, external third-party seals, service provider’s privacy policy documents, reputation, direct observations, and degree of compliance with laws or standards have been used instead [27]. Preserving behavioral privacy requires more sophisticated approaches than just removing direct identifiers (IP address, social security number (SSN), blurring a face) or intuitive quasi-identifiers (gender, age, ethnicity) from databases [72].

Consequently, a person today has just a few or no possibilities to maintain privacy in networked information systems, and therefore just a few reasons to trust. According to Goldberg, the service user can only use feelings or personal opinions as measures of the level of privacy and trust [39], reject the use of the service, filter the amount of PII he or she is willing to disclose, or add noise to data before disclosure [76]. This all indicates that the current situation is unsatisfactory.

## 5. Novel Approaches for Privacy and Trust

As discussed earlier, current privacy and trust models and solutions are insufficient to provide an acceptable level of privacy in networked and highly distributed information systems. Furthermore, a service user has few or no reasons to trust the service provider or the ecosystem as a whole. To solve these problems, researchers have created different privacy and trust approaches and solutions. Most of them focus on reducing the negative consequences of the use and sharing of personal information by offering more control and the possibility of using computer-understandable policies. There are also solutions, providing insight that the control model and the use of consent are inadequate and that a more radical solution is needed (Table 1).

The aim of the privacy as control approach is to give the DS or service users more control over what data they wish to share with whom and how and for what purposes the data can be used. Privacy nudges offer the person a ready-made template to make personal choices. On the other side, it is only a normative “one-size-fits-all” solution to make normative assumptions about the value of privacy [77]. User-tailored privacy solutions offer the user more flexibility by automatically tailoring IS’s privacy settings to fit the user’s privacy preferences [78]. As AI applications have the ability to predict a user’s privacy preferences by determining privacy needs based on the user’s previous data sharing history, AI can be used to contextually tailor a user’s privacy needs.

The person-controlled EHR (Electronic Health Record)/PHR (Personal Health Record) approach gives a person full control over his/her own PHI (e.g., the person grants or rejects granular access to the stored PHI in a context). Typically, rules that are expressed in the form of personal policies and data encryption methods are used together. The encrypted data can be stored on a blockchain [79]. Yue et al. have proposed a person-controlled blockchain solution that enables the patient to own, control, and share their own data securely without violating privacy [71].

Computational privacy models use mathematical methods to calculate the level of privacy using measured attributes and mathematical methods. According to Ruotsalainen et al., computational privacy offers a better approximation for the actual level of privacy than risk probabilities and privacy perceptions [48]. A contractual agreement, such as a legally binding service level agreement (SLA), is widely used between organizations. Ruotsalainen et al. have proposed the use of legally binding digital (Smart) contracts between the pHealth customer and service provider [27,80]. A smart contract is a set of rules that can be executed in a network of mutually distrusted nodes without the need for a centralized, trusted authority [81]. To guarantee the integrity, availability, and non-repudiation of the contract, it can be stored on a Blockchain. In a smart contract, the service user’s personal privacy policy is part of the contract between the person and the pHealth service provider. The personal policy can regulate not only how the service provider uses PHI but also the sharing and secondary use of PHI in the ecosystem [54].

New cryptographic solutions such as encryption, differential privacy, k-anonymity, and homomorphic encryption offer ways to maintain privacy. Homomorphic encryption allows some calculations with the data without decryption [82]. Architectural solutions such as edge-and-fog computing can also support privacy. The edge consists of human-controlled devices, such as PCs, smart phones, IoT devices, personal health devices, and local routers [83]. In edge computing, the processing of sensitive data takes place at the local level, and the Edge router controls the data flow between the edge domain and other worlds [84].

Some researchers have stated that the current privacy and trust models are unsatisfactory and that a radical (paradigmatic) change is necessary [54]. According to Ritter, today’s highly distributed information systems, such as ecosystems and the Internet of Things (IoT), have raised legal questions such as who is the owner of PHI and behavioral personal data by defining a new class of property by legislation [85]. He also noted that today’s defensive privacy laws should be expanded to support new contractual models such as smart contracts, and the consumer should have a veto right concerning privacy [85]. Another radical solution is to make the person the legal owner of his or her PHI. This informational property rights model gives the person the power and ability to define how and by whom PHI is used. According to Samuelson, the informational property rights model empowers individuals to negotiate with organizations and firms about how data is used [86]. Koos has proposed a variant of this model where the PHI can be licensed by the customer [87]. Ruotsalainen et al. have proposed PHI as a personal property model, where a person defines policies for the use and sharing of PHI in the ecosystem [27]. To be effective, the property model requires legal support [85]. The property model can also be expanded to cover a person’s behavioral data.

Trust creation by information and explanations regarding how information systems function seems to make information systems more trustworthy [88]. Challenges in this transparency model are the lack of reliable information about system trust features and the fact that explanations and increased information overload the user in a situation. To outweigh this, Ruotsalainen et al. have proposed for pHealth the use of a computational trust model that is based on information about the ecosystem’s measured/published features. In this solution, a Fuzzy Linguistic method is used to calculate the merit of service (fuzzy attractiveness rating) for the whole ecosystem [48]. Depending on the quality of the available attributes (i.e., attributes should be measurable if possible), this model can support the idea of building rational trust.

A radical approach is the use of the concept of informational duties instead of privacy. Information duties imply that individuals and institutions acting as data controllers or processors have specific information duties towards data subjects [89,90]. For privacy, Balkin has proposed the deployment of the concept of information fiduciary as a specific duty [67]. Fiduciaries must act in the interests of another person, i.e., a fiduciary has a responsibility to accept and act based on privacy needs expressed by a person. Fiduciaries also have obligations of loyalty and care toward another person and the responsibility not to do harm [67]. Therefore, according to Barret, the information fiduciary model has the power to strengthen equality and autonomy in the digital society and to offer better privacy protection [91]. According to Dobkin, the principle of the information fiduciary should be legally imposed as a duty in digital information systems [66].

The fiduciary relationship, as a legal duty, can also be used as a trust builder. According to Mayer, trust in fiduciary relationships is based on the professional’s competence and integrity [92]. Waldman sees that privacy in an information-sharing context is a social construct based on trust. He has proposed for privacy the privacy as trust model. According to Waldman, privacy as trust creates a fiduciary relationship between data subjects and users. In this approach, a private context is also a trusted context [38].

Blockchain technology can be used as a trust builder because it offers decentralized trust. In blockchain, people do not need interpersonal trust, but users must trust mathematics, algorithms, and indirectly, the creators of the blockchain system [93].

In an ecosystem, a single privacy or trust solution alone is hardly a silver bullet, and the combination of different methods shown in Table 1 offers a better solution. Ruotsalainen et al. have developed a solution that combines privacy as a personal property model, trust as a fiducial duty, a legally binding smart contract, and blockchain-based repositories for pHealth [27]. Thereby, the smart contract is a digital SLA agreement the service provider has a legal duty to follow.

As already mentioned, the information processing in the pHealth ecosystem should be ethically acceptable. Therefore, pHealth information systems should be compliant with the principles of information ethics (non-maleficence, beneficence, justice, and respect for autonomy). Hand has proposed a solution for an ethical information system that is based on the following ethical principles: integrity, honesty, objectivity, responsibility, trustworthiness, impartiality, nondiscrimination, transparency, accountability, and fairness [94].

## 6. A Holistic View to Privacy and Trust in pHealth Ecosystems

As discussed in earlier chapters, the authors expect that future pHealth will be part of a highly distributed and dynamic multi-stakeholder ecosystem, i.e., an information system that collects and shares all kinds of PHI and intensively uses AI, ML, and DL for detailed personal health analysis. In an ecosystem, some stakeholders can be virtual; PHI and results are shared not only between the user and the service provider but increasingly with other stakeholders across contexts and jurisdictions [2]. Furthermore, stakeholders in the pHealth ecosystem often have different business and privacy policies as well as trust features. To dare to use offered services, the user needs to know the aggregated level of privacy and reasons to trust not only a service provider but the ecosystem as a whole. In this chapter, the authors create a holistic solution to this challenge.

According to Holt et al., in modern highly distributed ecosystems, infrastructure, policies, citizen rights, national and international regulations and laws, as well as cultural preferences and corporate policies, make the maintenance of privacy and trust an extremely complex task [95]. Elrik has noted that ecosystem interrelations between members define how the ecosystem works, and a holistic approach to privacy is needed [96]. The authors state that in future pHealth ecosystems, privacy and trust cannot be built using a single method or solution. Instead, a holistic, systemic view is needed. Furthermore, for privacy and trust, a user-centric approach should be used [47].

For the future pHealth ecosystem, the authors have created a holistic, six-level, user-centric conceptual model (Figure 2). This model also supports the idea of explainability, i.e., that service users of the ecosystem should understand how their PHI is processed and used and how their privacy and trust needs are implemented by different stakeholders. The authors also expect that explainability fosters trust in the ecosystem [97].

For each level of the conceptual model, the authors have analysed from the user’s point of view the privacy and trust challenges and their possible solutions. The content of Table 2 creates a holistic view of privacy and trust in a future pHealth ecosystem.

Using the “Conceptual Model of Everyday Privacy in Ubicompo” developed by Lederer et al. [98], as a starting point, the authors have conceptualized the influence of privacy and trust factors discussed in Table 2 and developed the following formula:Level of privacy and trust = f (M, E, Te, IA, SR, SP, KN, USPr, USPt, DS) (1)
where

M = Models for ethics, privacy, and trust

E = Environment (e.g., Laws, regulations, standards) [98]

Te = technology (e.g., safeguards, encryption) [99]

IA = Information architecture [98,100]

SR = stakeholders’ privacy and trust features and their relations [48]

SP = service provider’s privacy and trust features (attributes) [48]

KN = knowledge [101]

USPr = service user’s privacy needs

USPt = service users trust requirements and trust threshold

DS = sensitivity of data [98].

Using the analysis made in the previous chapter and the content of Table 2, the authors have also developed a proposal for an ethically acceptable, trustworthy, and privacy-enabled pHealth ecosystem. Concerning the ethical model, the authors state that consequentialism (i.e., consequences to a person caused by the collection, use, and disclosure of PHI) alone is insufficient for future pHealth. Instead, the authors propose that a combination of consequentialism, duty ethics, and utilitarianism (i.e., the use of PHI should be available to improve the population’s health) should be used in the environment. Furthermore, the privacy as personal property model is proposed to be used. This implies that the DS or service user has legal ownership of their own PHI, including personal health behaviors.

As the service user in the pHealth ecosystem is fully dependent on service providers’ fairness and knowledge concerning privacy and trust features of the ecosystem (Chapter 3), the authors propose for the trust model a solution where trust is a legally binding fiducial duty. Thereby, the service provider and other stakeholders have the legal duty not to prioritize their own business benefits but to take into account service users privacy needs [2,27]. It is also necessary that the service provider publish not only the trust and privacy features of its own information system but also proof of accountability. For this purpose, the service provider must enable the service user to access the audit trail concerning the use of collected PHI. New laws are also needed to strengthen the position of the service user. First, a transparency law is inevitable, which enables the service user to know how and by whom his or her PHI is collected and used and to be aware of what behavioral health data is collected. Secondly, a law for privacy as informational property and a law that supports legal, binding smart collection contracts are needed. The service user should have the choice of a paid pHealth application without health data collection and behavioral tracking.

New information architectures, such as blockchain-based information systems and edge architecture, offer increased privacy compared with currently widely used Cloud platform solutions. Therefore, they are good candidates for future pHealth systems. Federated learning (FL) is another interesting architectural solution. Encryption as a default principle should be used everywhere where it is possible (e.g., homomorphic encryption or differential privacy).

## 7. A Proposal for Privacy Enabled and Trustworthy for pHealth Ecosystem

Based on previous analysis and the holistic view (Chapter 6), the authors have developed a proposal (an example) of how a high level of privacy and trust can be reached in the pHealth ecosystem. It is assumed that any person has the right to control themselves, including data that describes their own thoughts, behaviors, emotions, values, and personal health-related information [50], and privacy is understood as the amount of power a person has against others control and manipulation over them. The proposal is called a hybrid solution by the authors because it is not aimed at superseding current general privacy protection laws such as the EU GDPR and laws regulating the collection and use of clinical data in health care. The authors’ proposal combines PHI as personal property, trust as a fiduciary duty for the service provider and other stakeholders processing PHI in the ecosystem, and a legally binding smart contract that is stored in a blockchain-based repository [27,102]. Property is a special personal property that cannot be traded to any private organization and is not monetarized [103]. Property is an allocation of power to the DS to define what is a fair collection and use of data and how PHI can be used and shared. This power also enables the DS to exclude others [104]. The property should be supported by a new property law. Other elements in this proposal are transparency, edge architecture, data encryption at the sensor, and communication levels.

## 8. Discussion

Even though information privacy and a high level of trust are prerequisites for successful pHealth, researchers have shown that in ecosystems, current privacy approaches and solutions do not offer a level of privacy acceptable for the service user, so trust is just an illusion. This indicates that future pHealth cannot be built on current privacy models and technology [27]. Furthermore, even the most modern privacy laws, such as the EU-GDPR, rely on insufficient privacy as the notice and choice concept (aka consent model) and on risk analysis that is inadequate in a future pHealth environment [105,106]. Concerning trust, service users are widely expected to blindly trust companies’ promises [75]. This all means that, until today, policy makers and technology firms have failed to provide people with reasons to trust information systems and have left users of digital networks and services vulnerable. There are many new technical, architectural, and mathematical privacy and trust solutions, but the authors state that none of them alone is sufficient because privacy and trust in ecosystems are interconnected and holistic system problems.

In this paper, the authors have developed a user-centric five-level conceptual model for privacy and trust in the pHealth ecosystem and a formula for its privacy and trust factors. For each level of the model, the authors have analysed privacy and trust challenges and their possible solutions. The results are shown in the template to provide a holistic view of privacy and trust. The template and the formula have many use cases. The service user can use them to evaluate the level of privacy and trust in pHealth. Furthermore, the service provider can use the template and formula to assess what knowledge should be disclosed to the user. Finally, a developer of a pHealth information system can use the template to plan the required privacy and trust services.

The authors have also made proposals for a future ethical, trusted, and privacy-enabled pHealth ecosystem. Here, PHI and health-related behaviors and emotions are the service user’s/DS’s personal property. Therefore, the DS has the power to express its own privacy and trust needs to the service provider and other ecosystem stakeholders. Transparency and accountability make it possible for the service user to estimate the actual level of privacy and trust in the ecosystem. For estimating the level of trust, the authors propose the use of computational methods [48]. In the proposed solution, users’ privacy and trust needs are stored in the form of computer-understandable policies and smart contracts on the blockchain. The legal binding duty concept in the model (fiducial duty) can guarantee fair information processing in the whole ecosystem.

Nevertheless, there are also challenges to be solved. According to Notario et al., ethical concepts, privacy, and trust principles and laws are typically described using high-level terms, and it is difficult to translate them into technical requirements and to support service users concerns and expectations [107]. Some researchers have argued that blockchain technology does not need trust to operate because there is no centralized trust anchor. Instead, according to De Flilippi, blockchain technology produces confidence (and not trust) [108]. Trust in the blockchain is nevertheless necessary because users need trust in the developers and the implementation of algorithms, mathematical knowledge, and cryptographic tools. The authors state that organizational trust and trust in technology are essential elements of ecosystems. What is needed is a commonly accepted definition of the meaning of organizational trust [93].

A big challenge in the authors’ proposal is that it requires new internationally accepted regulations and laws. Nowadays, there is no guarantee that policy makers have the intention to enact necessary legislation and force big internet vendors to support and implement laws that can strongly impact their current money-making model. Instead, the responsibility to manage online privacy has been increasingly transferred to service users [109]. Another challenge is to make organizations understand that only systems that behave ethically can be trusted [39]. Some researchers have also argued that a law for data as personal property may be difficult to construct, and this kind of property approach will cause economic losses and less innovation. The authors see that the hybrid approach discussed in this paper is, from the service user’s/DS’s point of view, a more preferable solution than PHI as a public/common good and current laws.

The authors state that despite researchers’ efforts to develop innovative technological privacy solutions, they alone will hardly make future pHealth ecosystems ethical and trustworthy and guarantee information privacy. Instead, it is necessary to start the development of next-generation pHealth ecosystems using a holistic view and a system-theoretical, context-aware, architecture-centred, ontology-based, and policy-driven approach [110] as standardized in ISO 23903:2021 Health informatics: interoperability and integration reference architecture: model and framework [111]. Therefore, the privacy and trust approaches and solutions discussed in this paper will be deployed. It is also necessary to understand privacy and trust in ecosystems at the system level and create new laws to strengthen a person’s position. The new solution should also support transparency and explainability. In the long run, global harmonization of how privacy and trust are understood and international regulations are also needed. In agreement with Schneiderman, the authors state that the future pHealth ecosystem shall be created and operated to respect, promote, and protect internationally recognized human rights [112].

## Figures and Tables

**Figure 1 jpm-13-01048-f001:**
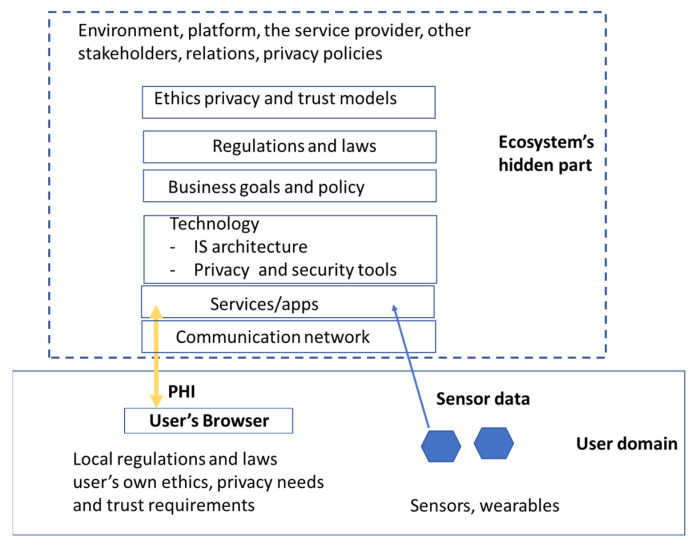
Users’ view on pHealth ecosystem.

**Figure 2 jpm-13-01048-f002:**
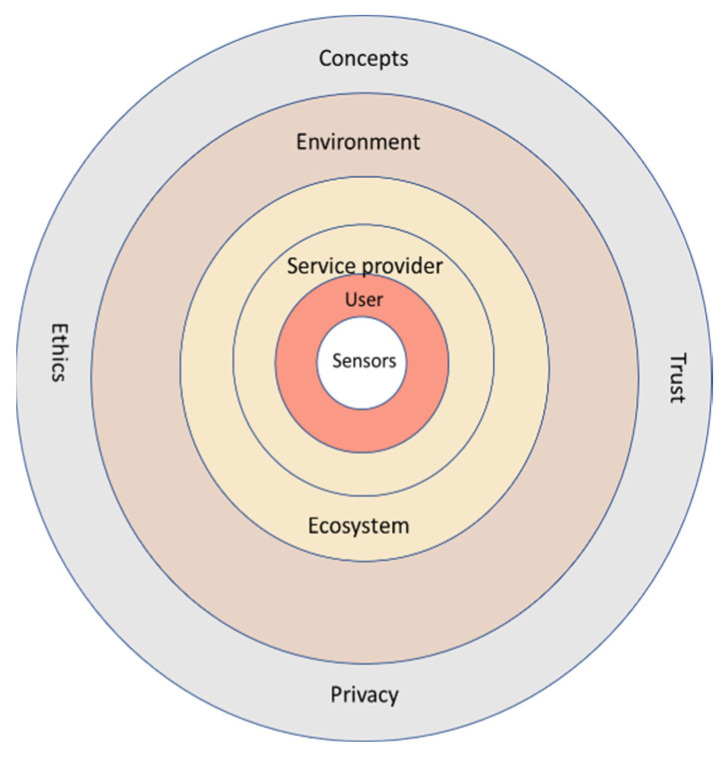
User centric five-level conceptual model for future pHealth ecosystem.

**Table 1 jpm-13-01048-t001:** Examples of new privacy and trust approaches and solutions.

Approach	Examples of Solutions
More personal control Transparency	Privacy nudges User tailored privacy Personalized privacy Person/Patient controlled or PHR Personal privacy policies Explainable trust
Ownership model	Privacy as intellectual property
Duty based model	Informational duties Trust as duty
Regulatory model	Trust as legal binding duty Accountability Privacy risk analysis
Computational models	Calculated level of privacy Calculated trust
Contractual models	Privacy negotiation Smart contract
Cryptographic based models	Blockchain Differential privacy Homomorphic encryption
Obfuscation methods Disclosure limitation	Data hiding by masking Adding noise or laying
Architectural solutions	Edge/Fog computing Federated learning
Ethics based approaches	Ethical design Ethical agents
Distributed trust approaches	Blockchain

**Table 2 jpm-13-01048-t002:** Holistic view to privacy and trust in a future pHealth ecosystem.

Levels	Content	Possible Solutions	Challenges
**Concepts and** **models**	Ethical model, principles and values Trust model Privacy model	Consequentialism Utility or duty ethics. Trust as informational duty Computational trust Privacy as property Personal tailored privacy	Stakeholders’ ethical models, values and principles are seldom known Stakeholders’ privacy and trust models used are not known Stakeholders do not do what they have promised in privacy documents Privacy and trust responsibilities are often unclear
**Environment**	Laws, standards and Golden Rules	New laws needed to: -Force transparency of privacy and trust features -Strengthen the role of person -Restrict hidden collection of the PHI	Ecosystem is highly distributed and cross-border Conflighting laws and privacy and trust models Laws should be global
**Ecosystem** *The service* *provider*	Stakeholders’ relations and privacy and trust features. ICT-architecture and technology Business model, Privacy policy Trust features of processes and applications	Transparency of business and privacy policies, stakeholders’ relations, and features. Edge and blockchainb architectures Federated computing Data encryption	Stakeholders’ business and policies vary. Stakeholder’s relations, privacy and trust features of information systems are not known Management of encryption keys Regulatory compliance and accountability. DS’s policy and stakeholser’s business policy can be conflighting Measurements of possible harm and the level of trust and privacy
**User/DS** **(Physical view)**	Users and the DS personal privacy and trust models. Expression of user’s privacy and trust needs	Personal privacy policies Tools to collect data and calculate the actual level of trust and privacy Evaluations of expected benefits and possible harms. Smart contracts Data encryption	No reason to trust Lack of: -Privacy and trust related data -Regulatory support -Practival tool for privacy management -Power to make contract ór negotiate No audit trails
**Data and** **sensors** **(perception)**	Raw data from sensors Self-disclosed PHI	Lite point-to point- encryption of data at sensor level	Data integrity, reliability and availability Lack of computational power for encryption

## Data Availability

Not applicable.

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
