# Peer review of "Future pHealth Ecosystem-Holistic View on Privacy and Trust"

_jpm, 2023, doi:10.3390/jpm13071048_

Round 1
Reviewer 1 Report
The paper is good
1. In abstract: it is too lengthy, optimize it
2. Add paper organization in introduction section
3. Is fig. 1 a drafted or taken from external source, provide reference if sourced.
4. Many paragraphs are uneven, make sure the uniform is followed, Minimum 7 lines per paragraph
Author Response
Dear Editor
Dear Review
Thank you for giving me the opportunity to submit a revised draft of my manuscript titled Future pHealth Ecosystem – Holistic View on Privacy and Trust (Manuscript ID jpm-2417361). We appreciate the time and effort that you and the reviewers have dedicated to providing your valuable feedback on my manuscript. We are grateful to the reviewers for their insightful comments on my paper. We have been able to incorporate changes to reflect most of the suggestions provided by the reviewers. We have highlighted the changes within the new version of the manuscript.
Here is a point-by-point response to the reviewers’ comments and concerns.
Comments from Reviewer 1
Comment 1: In abstract: it is too lengthy, optimize it
Response: Thank you for pointing this out. We have created a new shorter and optimized abstract that substitutes the original.
Modern pHealth is an emerging approach, collecting and using personal health information (PHI) for personalized healthcare and personalized health management. For its products and services, it deploys advanced technologies such as sensors, actuators, computers, mobile phones, etc. Researchers have shown that today’s networked information systems such as pHealth ecosystems miss appropriate privacy solutions, and trust is only an illusion. In the future, the situation will be even more challenging, because pHealth ecosystems will be highly distributed, dynamic, increasingly autonomous, multi-stakeholder eco-system with the ability to monitor the person’s regular life, movements, emotions and health related behaviour in real time. In this paper, the authors demonstrate that privacy and trust in ecosystems is a system level problem needing a holistic system-focused solution. To make future pHealth ethically acceptable, privacy-enabled and trustworthy, the authors have developed a conceptual five-level privacy and trust model as well as a formula that describes the impact of privacy and trust factors to the level of privacy and trust. Furthermore, the authors have analysed privacy and trust challenges and possible solutions at each level of the model. Based on the performed analysis, a proposal for future ethically acceptable, trustworthy and privacy-enabled pHealth is developed. The solution combines privacy as personal property and trust as legally binding fiducial duty approaches, and uses a Blockchain-based smart contract agreement to store person’s privacy and trust requirements and service providers’ promises.
Comment 2: Add paper organization in introduction section
Response: We agree. We have added paragraph that explains the organization of the paper to the end of introduction chapter
The rest of the article is organized as follows. Chapter 2 briefly summarizes main features of widely used privacy and trust models and principles of information ethics. In chapter 3, the authors define how pHealth ecosystem is understood in this paper and present user’s view to it. In chapter 4, privacy and trust challenges existing in current pHealth ecosystems are discussed. Then (chapter 5) features of new privacy and trust approaches developed by researchers are analysed. In chapter 6, a five-level holistic model and a formula describing factors which influence to level of privacy and trust in an ecosystem are presented. In chapter 7, the authors propose a holistic solution for trustworthy, privacy enabled and ethically acceptable pHealth ecosystem. Chapter 8 covers the limitations of this paper and outlines necessary future steps needed to reach the authors’ goal.
Comment 3: Is fig. 1 a drafted or taken from external source, provide reference, if sourced?
Response: Thank you for pointing this out. The Figure 1 is original and made by the authors just for this paper.
Comment 4: Many paragraphs are uneven make sure the uniform is followed, minimum 7 lines per paragraph.
Response: Thank you for pointing this out. Short paragraphs are reorganized where possible.
Additional clarifications
In addition to the above comments, spelling and grammatical errors found by the authors have been corrected.

Reviewer 2 Report
The article entitled “Future pHealth Ecosystem- Holistic View on Privacy and Trust” is well-written and, from my point of view, would be of interest for the readers of the Journal of Personalized Medicine. In spite of this and before its publication, I would like to suggest the following changes.
Nothing to say about the introduction as from my point of view it is well balanced.
About section “2. Privacy, Trust and Information Ethics” I suggest authors to enlarge section and give mode details about those references that are cited.
In section “3. User View on Privacy and Trust in pHealth Ecosystems”, at the beginning, it is said “An ecosystem is a dynamic system with abstract and technical elements”. If the definition was created by the authors they should say that from the point of view of authors….if not, the reference should be included in the statement.
In the first paragraph of “6. A Holistic View to Privacy and Trust in pHealth Ecosystems” a more in depht discusión of the thought of the authors should be performed.
From my point of view, Table 2 does not follow the format of the Journal.
English Language is fine.
